# Implicit Bias of SGD for Diagonal Linear Networks: a Provable Benefit of Stochasticity

Scott Pesme
EPFL
scott.pesme@epfl.ch

Loucas Pillaud-Vivien
EPFL
loucas.pillaud-vivien@epfl.ch

Nicolas Flammarion
EPFL
nicolas.flammarion@epfl.ch

## Abstract

Understanding the implicit bias of training algorithms is of crucial importance in order to explain the success of overparametrised neural networks. In this paper, we study the dynamics of stochastic gradient descent over diagonal linear networks through its continuous time version, namely stochastic gradient flow. We explicitly characterise the solution chosen by the stochastic flow and prove that it always enjoys better generalisation properties than that of gradient flow. Quite surprisingly, we show that the convergence speed of the training loss controls the magnitude of the biasing effect: the slower the convergence, the better the bias. To fully complete our analysis, we provide convergence guarantees for the dynamics. We also give experimental results which support our theoretical claims. Our findings highlight the fact that structured noise can induce better generalisation and they help explain the greater performances of stochastic gradient descent over gradient descent observed in practice.

## 1 Introduction

Understanding the performance of neural networks is certainly one of the most thrilling challenges for the current machine learning community. From the theoretical point of view, progress has been made in several directions: we have a better functional analysis description of neural networks [3] and we steadily understand the convergence of training algorithms [29, 10] as well as the role of initialisation [20, 12]. Yet there remain many unanswered questions. One of which is why do the currently used training algorithms converge to solutions which generalise well, and this with very little use of explicit regularisation [39].

To understand this phenomenon, the concept of *implicit bias* has emerged: if over-fitting is benign, it must be because the optimisation procedure converges towards some particular global minimum which enjoys good generalisation properties. Though no explicit regularisation is added, the algorithm is implicitly selecting a particular solution: this is referred to as the implicit bias of the training procedure. The implicit regularisation of several algorithms has been studied, the simplest and most emblematic being that of gradient descent and stochastic gradient descent in the least-squares framework: they both converge towards the global solution which has the lowest squared distance from the initialisation. For logistic regression on separable data, Soudry et al. show in the seminal paper [31] that gradient descent selects the max-margin classifier. This type of result has then been extended to neural networks and to other frameworks. Overall, characterising the implicit bias of gradient methods has almost always come down to unveiling mirror-descent like structures which underlie the algorithms.

35th Conference on Neural Information Processing Systems (NeurIPS 2021).

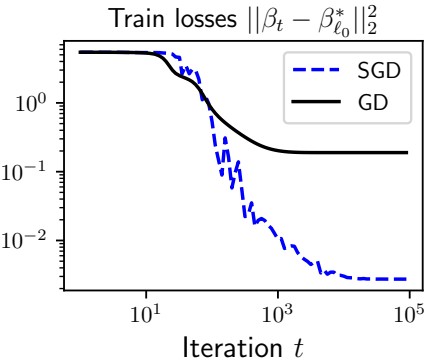
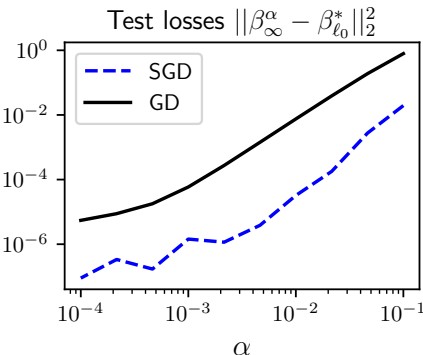

Figure 1: Sparse regression with $n = 40$, $d = 100$, $\|\beta_{\ell_0}^*\|_0 = 5$, $x_i \sim \mathcal{N}(0, I)$ $y_i = x_i^\top \beta_{\ell_0}^*$ . *Left*: for initialisation scale $\alpha = 0.05$, SGD converges towards a solution which generalises better than GD. *Right*: for different values of the initialisation scale $\alpha$, the solution recovered by SGD has better validation loss than that of GD. The sparsifying effect due to their implicit biases differ by more than an order of magnitude. See Section 5.1 for the precise experimental setup.

While mostly all of the results focus on gradient descent, it must be pointed out that this full batch algorithm is not used in practice for neural networks since it does not lead to solutions which generalise well [23]. Instead, results on stochastic gradient descent, which is widely used and shows impressive results, are still missing or unsatisfactory. This has certainly to do with the fact that grasping the nature of the noise induced by the stochasticity of the algorithm is particularly hard: it mixes properties from the model's architecture, the data's distribution and the loss. In our work, by focusing on simplified neural networks, we answer to the following fundamental questions: do SGD's and GD's implicit bias differ? What is the role of SGD's noise over the algorithm's implicit bias?

The simplified neural networks which we consider are diagonal linear neural networks; despite their simplicity they have become popular since they already enable to grasp the complexity of more general networks. Indeed, they highlight important aspects of the theoretical concerns of modern machine learning: the neural tangent kernel regime, the roles of over-parametrisation, of the initialisation and of the step size. For a regression problem where we assume the existence of an interpolating solution, we study stochastic gradient descent through its continuous version, namely stochastic gradient flow (SGF). Though the continuous modelling of SGD has not yet led to many fruitful results compared to the well studied gradient flow, we believe it is because capturing the essence of the stochastic noise is particularly difficult. It has generally been done in a non realistic and over simplified manner, such as considering constant and isotropic noise. In our work, we attach peculiar attention to the adequate modelling of the noise. Tools from Itô calculus are then leveraged in order to derive exact formulas, quantitative bounds and interesting interpretations for our problem.

## 1.1 Main contributions and paper organisation.

In Section 2, we start by introducing the setup of our problem as well as the continuous modelisation of stochastic gradient descent. Then, in Section 3, we state our main result on the implicit bias of the stochastic gradient flow. We informally formulate it here and illustrate it in Figure 1:

**Theorem 1** (Informal)**.** *Stochastic gradient flow over diagonal linear networks converges with high probability to a zero-loss solution which enjoys better generalisation properties than the one obtained by gradient flow. Furthermore, the speed of convergence of the training loss controls the magnitude of the biasing effect: the slower the convergence, the better the bias.*

Unlike previous works [14, 36], in addition to characterising the implicit bias effect of SGF, we also prove the convergence of the iterates towards a zero-loss solution with high-probability. To accomplish this, we leverage in Section 4 the fact that the iterates follow a stochastic continuous mirror descent with a time-varying potential. We support our results experimentally and validate our model in Section 5.

## 1.2 Related work

As recalled, implicit bias has a recent history that has been initiated by the seminal work [31] on max-margin classification with $\log$-loss for a linear setup and separable data. This work has been extended to other architectures, *e.g.* multiplicative parametrisations [14], linear networks [22] and more general homogeneous neural networks [27, 11]. In [36] the authors show that the scale of the initialisation leads to an interpolation between the neural tangent kernel regime [20, 12] (which is a linear regression on fixed features) leading to $\ell_2$ minimum norm solutions and the rich regimes leading to $\ell_1$ minimum norm solutions. Note that these works focus on full batch gradient descent (or flow) and are deeply linked to mirror descent.

While the links between SGD's stochasticity and generalisation have been looked into in numerous works [28, 21, 16, 18, 24], no such explicit characterisation of implicit regularisation have ever been given. It has been empirically observed that SGD often outputs models which generalise better than GD [23, 21, 16]. One suggested explanation is that SGD is prone to pick flatter solutions than GD and that bad generalisation solutions are correlated with sharp minima, i.e., with strong curvature, while good generalisation solutions are correlated with flat minima, i.e., with low curvature [17, 23]. This idea has been further investigated by adopting a random walk on random landscape modelling [18], by suggesting that SGD's noise is smoothing the loss landscape, thus eliminating the sharp minima [24], by considering a dynamical stability perspective [38] or by interpreting SGD as a diffusion process [16, 21, 8]. Recently, label-noise has been shown to influence the implicit bias of SGD, by biasing the solution towards the origin for quadratically-parameterized models [15] or by implicitly regularising the expected squared norm of the gradient of the model with respect to the weights [5]. Thus, if the notion of implicit bias of GD is fairly well understood both in the cases of regression and classification, it remains unclear for SGD, and its explicit characterisation is missing.

The linear diagonal neural networks we consider have been studied in the case of gradient descent [33] and stochastic gradient descent with label noise [15]. In both cases the authors show that this model has the ability to implicitly bias the training procedure to help retrieve a sparse predictor. The link between gradient descent and mirror descent for this model has been initiated by [13] and further exploited by the same author in [37, 34] for its sparse inducing property.

Contrary to the deterministic case, the modelling of stochastic gradient descent as a stochastic differential equation is quite recent, see [28, 21]. However, as highlighted by [1], early attempts often suffer from the drawback that they model the noise using a constant covariance matrix. On the contrary, state dependant noise has now become the legitimate manner for modelling SGD as a stochastic gradient flow and it is shown in [26] that it can be done consistently. Yet, noise modelling still remains the principal issue [35] as it influences largely the behaviour of the dynamics [8, 9].

## 1.3 Notations

For input data $(x_1, \ldots, x_n) \in (\mathbb{R}^d)^n$ and output $(y_1, \ldots, y_n) \in \mathbb{R}^n$, we denote respectively $X \in \mathbb{R}^{n \times d}$ the design matrix whose $i$-th row is feature $x_i \in \mathbb{R}^d$ and $y \in \mathbb{R}^n$ the vector of outputs. $\mathbb{R}_+^*$ denotes the set of strictly positive real numbers. For $p = 1, 2$, the $\ell_p$-norm of $x \in \mathbb{R}^d$ is $\|x\|_p^p = \sum_i^d |x_i|^p$. The operations $\odot$ will stand for coordinate-wise product between vector: $[u \odot v]_i = u_i v_i$ and $u^2 = u \odot u$. For $p \in \mathbb{N}^*$, we also define $u^p := u \odot \ldots \odot u$, the $p$ times product of $u$ with itself. All inequalities between vectors should be understood value by value. For $f, g \in \mathbb{R}$, the existence of $C > 0$ such that $f \leq Cg$ and $Cg \leq f$ will be denoted $f \leq O(g)$ and $\Omega(g) \leq f$ respectively. We shall use the symbole $\widetilde{O}$ when this is true up to $\log$ factors. For a vector $u \in \mathbb{R}^d$, $\mathrm{diag}(u)$ denotes the $d \times d$ diagonal matrix which has its diagonal equal to $u$. For a matrix $M \in \mathbb{R}^{d \times d}$, $\mathrm{diag}(M)$ denotes the vector $(M_{11}, \ldots, M_{dd}) \in \mathbb{R}^d$. The indexed vector $\beta^*$ will stand for any $\beta$ interpolating the data, i.e. any vector in the affine space $\{\beta \in \mathbb{R}^d \ s.t, \ X\beta = Y\}$ of dimension at least $d - n$. Out of all these, let $\beta_{\ell_1}^* = \underset{\beta \in \mathbb{R}^d \ \text{s.t.} \ X\beta = y}{\arg\min} \|\beta\|_1$. For $z$ any vector, $z_\infty$ or $z^\infty$ will always designate of $\underset{t \to \infty}{\lim} z_t$.

## 2 Setup and preliminaries

### 2.1 Architecture and algorithm.

**Overparametrised noiseless regression.** We consider a linear regression problem with outputs $(y_1, \ldots, y_n) \in \mathbb{R}^n$ and inputs $(x_1, \ldots, x_n) \in (\mathbb{R}^d)^n$. We study an overparametrised setting $(n < d)$

and assume that there exists at least one interpolating parameter $\beta^* \in \mathbb{R}^d$ which perfectly fits the training set, i.e. $y_i = \langle \beta^*, x_i \rangle$ for all $1 \leq i \leq n$. We parametrise the regression vector $\beta$ as $\beta_w$ with $w \in \mathbb{R}^p$. We will see that though in the end our final models $x \mapsto \langle \beta_w, x \rangle$ are classical linear models whatever the parametrisation $w \mapsto \beta_w$, the choice of this parametrisation has crucial consequences on the solution recovered by the learning algorithms. We study the quadratic loss and the overall loss is written as:

$$L(w) = L(\beta_w) := \frac{1}{4n} \sum_{i=1}^{n} (\langle \beta_w, x_i \rangle - y_i)^2 = \frac{1}{4n} \sum_{i=1}^{n} \langle \beta_w - \beta^*, x_i \rangle^2,$$

where by abuse of notation we use $L(w) = L(\beta_w)$.

**2-layer diagonal linear network.** The simplest parametrisation of $\beta_w$ is to consider $\beta_w = w$ which corresponds to the classical least-squares framework. It is well known that in this case, many first order methods (GD, SGD, with and without momentum) will converge towards the same solution: we say that they have the same implicit bias. This is experimentally not the case for neural networks where SGD has been shown to lead to solutions which have better generalisation properties compared to GD [23]. To theoretically confirm this observation, we study a simple non-linear parametrisation: $\beta_w = w_+^2 - w_-^2$ with $w = [w_+, w_-]^\top \in \mathbb{R}^{2d}$. We point out that it is 2-positive homogeneous and that it is equivalent to the parametrisation $\beta_{u,v} = u \odot v$ with $u, v \in \mathbb{R}^d$. It should be thought of a simplified linear network of depth 2 (see [36, Section 4] for more details). We consider two weight vectors $w_+$ and $w_-$ (and not only $\beta_w = w^2$) in order to ensure that our final linear predictor parameter $\beta_w$ can take negative values. For the sake of completeness, the study of diagonal linear networks of arbitrary depth $p \geq 3$ is done in Appendix E.2. Also note that additionally to being a toy neural model, it has received recent attention for its practical ability to induce sparsity [33, 34, 15] or to solve phase retrieval problems [37].

**Stochastic Gradient Descent.** With this quadratic parametrisation, the loss now rewrites as: $L(w) = \frac{1}{4n} \sum_{i=1}^{n} \langle w_+^2 - w_-^2 - \beta^*, x_i \rangle^2$. Note that despite its simplicity, this loss is non convex and its minimisation is non trivial. The algorithm we shall consider is the well known SGD algorithm, where for a step size $\gamma > 0$:

$$\begin{aligned} w_{t+1,+} &= w_{t,+} - \gamma \langle \beta_w - \beta^*, x_{i_t} \rangle \, x_{i_t} \odot w_{t,+} \\ w_{t+1,-} &= w_{t,-} + \gamma \langle \beta_w - \beta^*, x_{i_t} \rangle \, x_{i_t} \odot w_{t,-} \end{aligned} \qquad \text{where } i_t \sim \text{Unif}(1, n). \qquad (1)$$

It is convenient to rewrite this recursion as

$$w_{t+1,\pm} = w_{t,\pm} - \gamma \nabla_{w_\pm} L(w_t) \pm \gamma \, \text{diag}(w_{t,\pm}) X^\top \xi_{i_t}(\beta_t), \qquad (2)$$

where $\xi_{i_t}(\beta) = -\big( \langle \beta - \beta^*, x_{i_t} \rangle \mathbf{e}_{i_t} - \mathbb{E}_{i_t} \big[ \langle \beta - \beta^*, x_{i_t} \rangle \mathbf{e}_{i_t} \big] \big) \in \mathbb{R}^n$ is a zero-mean *multiplicative* noise which vanishes at any global optimum ($\mathbf{e}_i$ denotes the $i^{\text{th}}$ element of the canonical basis). We point out that all the results we shall give hold for any initialisation such that $w_{t=0,+} = w_{t=0,-} \in \mathbb{R}^d$, under which we have that $\beta_{w_{t=0}} = 0$. To understand under what conditions the SGD procedure converges and towards which point it does, we shall consider its continuous counterpart which has the advantage of leading to clean and intuitive calculations. We highlight the fact that we consider a bath-size equal to 1 for clarity, however all our analysis holds for mini-batch SGD (with and without replacement) simply by considering an effective step-size $\gamma_{\text{eff}}$ instead of $\gamma$, this is clearly explained in Appendix A.

## 2.2 Stochastic gradient flow

Continuous time modelling of sequential processes offer a large set of tools, such as derivation, which come in helpful to understand the dynamics of the processes. This has led to a large part of the recent literature to consider continuous gradient flow in order and understand the behaviour of gradient descent on complicated architectures such as neural nets. However, the continuous time modelling of stochastic gradient descent is more challenging: it requires to add on top of the gradient flow a diffusion term whose covariance matches the one of SGD. Hence, it is fundamental to understand its structure and scale.

**Understanding the noise's structure.** As seen in equation (2), evaluated at $w_\pm$, the stochastic noise $\gamma \operatorname{diag}(w_\pm) X^\top \xi_{i_t}(w)$ has two main characteristics which we want to preserve:

- It belongs to $\operatorname{span}(w_\pm \odot x_1, \ldots, w_\pm \odot x_n)$
- It has covariance $\Sigma_{\text{SGD}}(w_\pm) := \gamma^2 \operatorname{diag}(w_\pm) X^\top \operatorname{Cov}_{i_t}(\xi_{i_t}(\beta)) X \operatorname{diag}(w_\pm) \in \mathbb{R}^{d \times d}$

It remains to understand the structure of the covariance of $\xi_{i_t}$ which has the following closed form: $\operatorname{Cov}_{i_t}(\xi_{i_t}(\beta)) = \frac{1}{n} \operatorname{diag}(\langle \beta - \beta^*, x_i \rangle^2)_{1 \le i \le n} - \frac{1}{n^2}\big(\langle \beta - \beta^*, x_i \rangle \langle \beta - \beta^*, x_j \rangle\big)_{1 \le i, j \le n}$. We identify the two key facts: (i) it is diagonal at the leading $n^{-1}$ order and (ii) its trace is linked to the loss as $\operatorname{Var}_{i_t}(\|\xi_{i_t}(\beta)\|_2) = \frac{4}{n} L(\beta) + O(\frac{1}{n^2})$. This leads us in modelling $\xi_{i_t}(\beta)$'s covariance matrix as $\frac{4}{n} L(\beta) I_n$ as it preserves these two characteristics [1]. Finally this brings us to consider the following modelling of the overall noise's structure: $\Sigma_{\text{SGD}}(w_\pm) \cong \frac{4}{n} \gamma^2 L(w) [\operatorname{diag}(w_\pm) X^\top]^{\otimes 2}$.

**Stochastic differentiable equation modelling.** Guided by the previous considerations, we study the following stochastic gradient flow:

$$
\begin{aligned}
\mathrm{d}w_{t,+} &= -\nabla_{w_+} L(w_t)\, \mathrm{d}t + 2\sqrt{\gamma n^{-1} L(w_t)}\, w_{t,+} \odot [X^\top \mathrm{d}B_t] \\
\mathrm{d}w_{t,-} &= -\nabla_{w_-} L(w_t)\, \mathrm{d}t - 2\sqrt{\gamma n^{-1} L(w_t)}\, w_{t,-} \odot [X^\top \mathrm{d}B_t],
\end{aligned}
\tag{3}
$$

where $\mathrm{d}B_t$ is a standard $\mathbb{R}^n$ Brownian motion. The SDE is a perturbed gradient flow with a diffusion term that is defined such that its Euler discretisation with step size $\gamma$ leads to a Markov Chain whose covariance exactly matches SGD's noise covariance $\Sigma_{\text{SGD}}(w_\pm)$. We refer to [26] or [25] for the technical details regarding consistency of such a procedure in the limit of small step sizes. This stochastic differential equation is the starting point of the analysis.

## 3 The implicit bias of the stochastic gradient flow

**Implicit bias and hyperbolic entropy.** To understand the relevance of the main result and how stochasticity induces a preferable bias, we start by recalling some known results for gradient flow. In [36] it is shown, assuming global convergence, that the solution selected by the gradient flow initialised at $\alpha \in \mathbb{R}^d$ and denoted $\beta_\infty^\alpha$ solves a constrained optimisation problem involving the *hyperbolic entropy* introduced by [13]:

$$
\beta_\infty^\alpha = \underset{\beta \in \mathbb{R}^d \text{ s.t. } X\beta = y}{\arg\min} \phi_\alpha(\beta) := \frac{1}{4} \Big[ \sum_{i=1}^d \beta_i \operatorname{arcsinh}\Big(\frac{\beta_i}{2\alpha_i^2}\Big) - \sqrt{\beta_i^2 + 4\alpha_i^4} \Big],
\tag{4}
$$

Though the hyperbolic entropy function has a non-trivial expression, its principal characteristic is that it interpolates between the $\ell_1$ and the $\ell_2$ norms according to the scale of $\alpha$. More precisely for $\alpha \in \mathbb{R}$ [2]: $\phi_\alpha(\beta) \underset{\alpha \to 0}{\sim} \frac{1}{2} \ln\big(\frac{1}{\alpha}\big) \|\beta\|_1$ and $\phi_\alpha(\beta) \underset{\alpha \to +\infty}{=} -\frac{1}{2}\alpha^2 + \frac{1}{16\alpha^2} \|\beta\|_2^2 + o(\alpha^{-2})$. We refer to [36, Theorem 2] for more details on the asymptotic analysis. The implicit optimisation problem (4) therefore highlights the fact that the initialisation scale of the weights controls the shape of the recovered solution. Small initialisations lead to low $\ell_1$-norm solutions which are known to induce good generalisation properties: this is what is often referred to as the *rich regime*. Large initialisations lead to low $\ell_2$-norm solutions: this is referred to as the *kernel regime* or *lazy regime* in which the weights move only very slightly. The dynamics of the gradient flow are then very similar to the one of kernel linear regression with the kernel depending on the initialisation [20, 12]. Overall, to retrieve a sparse solution, one should initialise with the smallest $\alpha$ possible. However, as is clearly explained in [36], it is important to stress out that there is a generalisation / optimisation tradeoff: the point $w = 0$ happens to be a saddle point for the loss and a smaller $\alpha$ will lead to a longer training time.

**Main result.** In the main theorem we show that, for an initialisation scale $\alpha$, the stochasticity of SGF biases the flow towards solutions which still minimise the hyperbolic entropy. However, what is remarkable is that it does so with an effective parameter $\alpha_\infty$ which is strictly smaller than $\alpha$. The recovered solution therefore minimises an optimisation problem which has better sparsity inducing properties than that of gradient flow.

---

[1] the general case is discussed in Appendix E.1

[2] If $\alpha \in \mathbb{R}$ we consider the abuse of notation $\phi_\alpha := \phi_{\alpha \mathbf{1}}$.

**Theorem 1.** *For $p \leq \frac{1}{2}$ and $w_{0,\pm} = \alpha \in (\mathbb{R}_+^*)^d$, let $(w_t)_{t \geq 0}$ follow the stochastic gradient flow* (3) *with step size $\gamma \leq O\big(\big[\ln(\frac{4}{p})\lambda_{\max} \max\{\|\beta_{\ell_1}^*\|_1 \ln\big(\frac{\|\beta_{\ell_1}^*\|_1}{\min_i \alpha_i^2}\big), \|\alpha\|_2^2\}\big]^{-1}\big)$ where $\beta_{\ell_1}^* = \underset{\beta \in \mathbb{R}^d \ s.t. \ X\beta = y}{\arg\min} \|\beta\|_1$ and $\lambda_{\max}$ is the largest eigenvalue of $X^\top X/n$. Then, with probability at least $1-p$:*

- *$(\beta_t)_{t \geq 0}$ converges towards a zero-training error solution $\beta_\infty^\alpha$*

- *the solution $\beta_\infty^\alpha$ satisfies*

$$\beta_\infty^\alpha = \underset{\beta \in \mathbb{R}^d \ s.t. \ X\beta = y}{\arg\min} \phi_{\alpha_\infty}(\beta) \quad where \quad \alpha_\infty = \alpha \odot \exp\left(-2\gamma \operatorname{diag}\left(\frac{X^\top X}{n}\right)\int_0^{+\infty} L(\beta_s)\,\mathrm{d}s\right). \quad (5)$$

The theorem is three-fold: with high probability and for an explicit choice of constant step size $\gamma$, (i) the flow $(\beta_t)_{t \geq 0}$ converges, (ii) its limit $\beta_\infty^\alpha$ is an interpolating solution, i.e. $X\beta_\infty^\alpha = y$, (iii) this solution minimises the hyperbolic entropy problem with a parameter that depends on the dynamics. We illustrate these results in Figure 2. Now let us comment further the theorem.

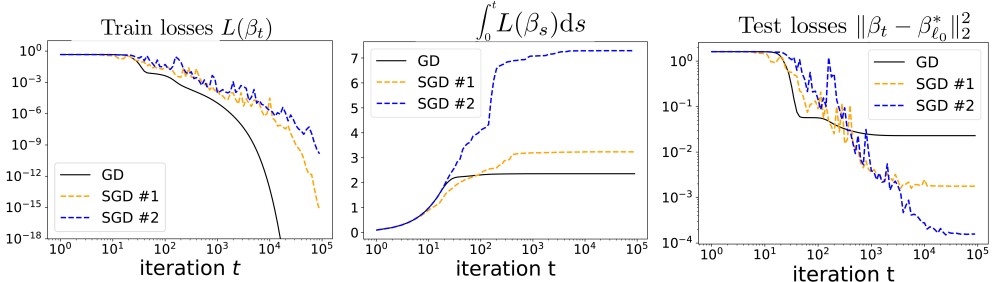

Figure 2: Sparse regression (see Section 5.1 for the detailed experimental setting). Both SGD and GD are initialised at $\alpha = 0.1$. 2 different runs of SGD over the training set are performed, they differ due to the inner stochasticity of the algorithm. *Left*: GD and SGD both converge towards a global minimum. *Middle and right*: for two different trajectories of SGD, the higher the value of the loss integral at convergence, the better the validation loss. In both cases SGD converges towards a solution which generalises better than GD. This figure illustrates Theorem 1.

**Beneficial implicit bias through effective initialisation.** The most remarkable aspect of the result is that the recovered solution $\beta_\infty^\alpha$ minimises the same potential as for gradient flow but with an *effective parameter* $\alpha_\infty$ which is strictly smaller than $\alpha$. Hence, the hyperbolic entropy is closer to the $\ell_1$ norm compared to the deterministic case, proving a systematic benefit of stochasticity. Note that this effective parameter is random and controlled by the loss integral $\int_0^{+\infty} L(\beta_s)\,\mathrm{d}s$: the higher the integral, the smaller the effective initialisation scale. In other words and quite surprisingly, the slower the loss converges to 0, the "richer" the implicit bias. However, it must be kept in mind that, as explained in [36], there is a tension between generalisation and optimisation: a longer training time might improve generalisation but comes at the cost of... a longer training time. Yet it is clear experimentally that SGD systematically largely wins the trade-off over GD (see Figure 2). Interestingly, Problem (5) tells us that the implicit bias of SGD initialised at $\alpha$ acts as if we run GD initialised at $\alpha_\infty$ (see Section 5.3). Note that the minimisation problem (5) only makes sense *a posteriori* since the quantity $\alpha_\infty$ depends on the whole stochastic trajectory. Finally, an interesting question is whether one can quantify the scale of this beneficial phenomenon, i.e. how small $\alpha_\infty$ is compared to $\alpha$. To answer this, we quantify the scale of the loss integral w.r.t. $\gamma$ and $\alpha$ (see Proposition 3) and show under slightly stronger conditions that the relative scale $\alpha_\infty/\alpha$ decays as power of $\alpha$ (See Eq. (8) of the main text and Proposition 6 of the appendix for a proof).

**Kernel regime.** Though it is less our focus, our result still holds as $\alpha \to +\infty$ which corresponds to the kernel regime. In this regime, we believe that $\int_0^{+\infty} L(\beta_s)\,\mathrm{d}s \underset{\alpha \to \infty}{\to} 0$ (not shown in the paper but experimentally observed) and hence SGF and GF converge towards the same solution. This is expected since in the NTK regime, the iterates follow a kernel linear regression for which the bias of SGF and GF are the same.

**Step size.** Note that the convergence of the iterates holds for a constant step size. This is not illogical since in the overparametrised setting, the noise vanishes at the optimum (see [32] for a convergence result in the overparametrised least-squares setup). The explicit formula for the $\gamma$ upper bound is $\gamma \leq \left( 400 \ln\left(\frac{4}{p}\right) \lambda_{\max}\left(\frac{X^\top X}{n}\right) \max\left\{ \|\beta_{\ell_1}^*\|_1 \ln\left(\sqrt{2}\frac{\|\beta_{\ell_1}^*\|_1}{\min_i \alpha_i^2}\right), \|\alpha\|_2^2 \right\} \right)^{-1}$. It has a classical dependence on $\lambda_{\max}(X^\top X/n)$ which can be computed, but also on the unknown value of $\|\beta_{\ell_1}^*\|_1$. However in practice we choose the highest value of $\gamma$ for which the iterates converge. Note that in practice the weights are often initialised such that $\|\alpha\|_2^2$ is roughly equal to 1 and hence it is sensible to consider $\|\alpha\|_2^2 < \|\beta_{\ell_1}^*\|_1$. In the explicit bound, there is a $\ln\left(\|\beta_{\ell_1}^*\|_1/\min_i \alpha_i^2\right)^{-1}$ factor, we believe that it is an artefact of our analysis and could be removed. It is hence best to think of the upperbound on $\gamma$ to simply be $\gamma \leq O(\frac{1}{\lambda_{\max}\|\beta_{\ell_1}^*\|_1})$.

**Convergence and proof sketch.** Let us put emphasis on the fact that since we deal with a non-convex problem, neither convergence nor convergence towards a global minimum are obvious. In most of similar works, convergence of the iterates is assumed [36, 14]. In fact, the hardest and most technical part of our result is to show the convergence of the flow with high probability: once the convergence is shown, describing the minimisation problem $\beta_\infty^\alpha$ verifies is straightforward. In the following section we give several properties which constitute the major keys of the theorem's proof.

## 4 Links with mirror descent

The aim of this section is to show that the sequence $(\beta_t)_{t\geq 0}$ follows a stochastic version of continuous mirror descent with a time dependent mirror. From this crucial property, we show how the convergence and implicit bias characterisation follow. Finally, as it is one of the central objects of our main theorem, we give an estimation of $\int_0^\infty L(\beta_s) \, ds$.

### 4.1 Stochastic continuous mirror descent with time-varying potential

We start by recalling known results on the link between implicit bias and mirror descent. We recall also convergence guarantees for mirror descent dynamics.

**Mirror descent: convergence and implicit bias.** For any $\beta_0 \in \mathbb{R}^d$ and convex potential function $\Psi$, consider the mirror descent flow $(\beta_t)_t$ which corresponds to $d\nabla\Psi(\beta_t) = -\nabla L(\beta_t)dt$. Though the convergence of the loss to $0$ is straightforward, showing the convergence of the iterates requires more work and is shown in [4, Theorem 2] for strongly convex potentials. Yet, once the convergence of the iterates is shown, deriving the implicit minimisation problem is straightforward. We recall the reasoning here (see Section 3 of [2] for more details): integrating the flow yields $\nabla\Psi(\beta_\infty) - \nabla\Psi(\beta_0) = -\int_0^\infty \nabla L(\beta_s) \, ds = -4X^\top \int_0^\infty X(\beta_s - \beta_\infty) \, ds \in \mathrm{span}(X)$. This condition, along with the fact that $X\beta_\infty = y$ exactly corresponds to the KKT conditions of the problem:

$$\beta_\infty = \underset{\beta \in \mathbb{R}^d \text{ s.t. } X\beta = y}{\arg\min} \ D_\Psi(\beta, \beta_0), \tag{6}$$

where $D_\Psi(\beta, \beta_0) = \Psi(\beta) - \Psi(\beta_0) - \langle \nabla\Psi(\beta_0), \beta - \beta_0 \rangle$ is the Bregman divergence w.r.t. $\Psi$.

**Link with our model.** It turns out that these general observations on mirror descent apply to our framework when $(w_t)_t$ follows the gradient flow $dw_{t,\pm} = -\nabla_{w_\pm} L(w_t) \, dt$. Indeed it has been shown in [36] that the corresponding iterates $\beta_t = w_{t,+}^2 - w_{t,-}^2$ follow a mirror descent with potential $\phi_\alpha$ defined in Eq.(4). Therefore we can apply the previous remarks to obtain the convergence towards an interpolator[3], as well as the associated implicit minimisation problem which in our case can be rewritten as $\beta_\infty^\alpha = \underset{\beta \in \mathbb{R}^d \text{ s.t. } X\beta = y}{\arg\min} \ \phi_\alpha(\beta)$ since $\nabla\phi_\alpha(\beta_0 = 0) = 0$.

**Stochastic Mirror descent with a time varying potential.** To address the problem where $(w_t)_t$ follows a stochastic gradient flow instead of a gradient flow, it is natural, as in the deterministic framework, to see what type of flow $(\beta_t)_t$ follows. Because of the noise, we cannot hope to simply

---

[3]In our case, $\phi_\alpha$ is not strongly convex so a bit more work is necessary to show the convergence of the iterates (see Appendix C).

recover a classical mirror descent. However interestingly the next property shows that it follows a stochastic mirror-like descent with a geometry that depends on time.

**Proposition 1.** *Consider the iterates* $(w_t)_{t \geq 0}$ *issued from the stochastic gradient flow in Eq.(3) with initialisation* $w_{0,\pm} = \alpha \in (\mathbb{R}_+^*)^d$. *Then the corresponding flow* $(\beta_t)_{t \geq 0}$ *follows a "stochastic continuous mirror descent with time varying potential" defined by:*

$$\mathrm{d}\nabla\phi_{\alpha_t}(\beta_t) = -\nabla L(\beta_t)\,\mathrm{d}t + \sqrt{\gamma n^{-1}L(\beta_t)}X^\top \mathrm{d}B_t, \tag{7}$$

*where* $\alpha_t = \alpha \odot \exp\left(-2\gamma\,\mathrm{diag}\left(\frac{X^\top X}{n}\right)\int_0^t L(\beta_s)\,\mathrm{d}s\right)$ *and* $\phi_\alpha$ *is the hyperbolic entropy defined in* (4).

Under this form we clearly see that the iterates $(\beta_t)_t$ follow a flow which closely resembles that of mirror descent but with two major differences: (i) the potential $\phi_{\alpha_t}$ changes over time according to the random quantity $\int_0^t L(\beta_s)\,\mathrm{d}s$, (ii) the flow is perturbed by noise. We highlight the fact that viewing the dynamics this way has the major advantage of giving a clear roadmap for the proof of Theorem 1: (i) we can adapt classical mirror-descent results to our framework and construct appropriate Lyapunov functions to prove the convergence of the flow with high probability to some interpolator $\beta_\infty^\alpha$, (ii) we immediately recover the corresponding minimisation problem as in the deterministic case. Indeed, integrating Eq.(7) still yields $\nabla\phi_{\alpha_\infty}(\beta_\infty^\alpha) \in \mathrm{span}(X)$ which, along with $X\beta_\infty^\alpha = y$, are the KKT conditions of the implicit minimisation problem (5). We emphasise the fact that the structure of the noise, belonging to $\mathrm{span}(X)$, is crucial in order to obtain this minimisation problem. This would for instance clearly not be true if we considered isotropic noise in the SDE modelling. This highlights the fact that not every form of noise improves the implicit bias: the shape of the intrinsic SGD noise is of primal importance [15].

## 4.2   Convergence and control of $\int_0^\infty L(\beta_s)\,\mathrm{d}s$

Though it seems easy to derive the implicit minimisation problem (5) from the mirror-like structure of Eq.(7), it is necessary to ensure that the iterates converge towards an interpolator $\beta_\infty$. This is the purpose of the following proposition.

**Proposition 2** (Convergence of the iterates). *Consider the iterates* $(w_t)_{t \geq 0}$ *issued from the stochastic gradient flow* (3), *initialised at* $w_{0,\pm} = \alpha \in (\mathbb{R}_+^*)^d$. *For* $p \leq \frac{1}{2}$ *and* $\gamma$ *such as in Theorem 1, then with probability at least* $1 - p$, *the flow* $(\beta_t)_t$ *converges to an interpolating solution* $\beta_\infty^\alpha$.

The convergence of the iterates is technical and requires several intermediate results. We start by considering an appropriate Bregman-type stochastic function with a time-varying potential and show that it converges with high probability. Leveraging the fact that we are able to bound the iterates $\beta_t$, we are able to show that the limit of the function is in fact 0. Owing to the fact that the function we consider also controls the distance of $\beta_t$ to a particular $\beta^*$ we finally get that the iterates converge.

However for the objects (such as $\alpha_\infty$) and functions we introduce to be well defined, we need to guarantee the convergence of $\int_0^\infty L(\beta_s)\mathrm{d}s$. Besides, it is crucial to grasp the scale of this quantity since it gives the overall scale of $\alpha_\infty$. This is done in the following proposition where we lower and upper bound its value.

**Proposition 3.** *Under the same setting as in Proposition 2 with initialisation* $w_{0,\pm} = \alpha\mathbf{1}$, *we have with probability at least* $1 - p$:

$$\Omega\left(\|\beta_{\ell_1}^*\|_1 \ln\left(\frac{\|\beta_{\ell_1}^*\|_1}{\alpha^2}\right)\right) \underset{\alpha \to 0}{\lesssim} \int_0^{+\infty} L(\beta_s)\,\mathrm{d}s \leq O\left(\max\left\{\|\beta_{\ell_1}^*\|_1 \ln\left(\frac{\|\beta_{\ell_1}^*\|_1}{\alpha^2}\right), \alpha^2 d\right\}\right).$$

We point out that the lower bound is given for small $\alpha$'s for simplicity but we provide in Lemma 7 (Appendix B.5) a lower bound which holds for all $\alpha$'s. Note that when $\gamma = 0$, which corresponds to deterministic gradient flow, we can give the exact value for the integral: $\int_0^{+\infty} L(\beta_s)\,\mathrm{d}s = \frac{1}{2}D_{\phi_\alpha}(\beta_\infty^\alpha, \beta_0)$ (see Proposition 7 in Appendix C). This matches the scale of the bounds given in Proposition 3, hence showing the tightness of the result. We focus now on how this translates to the scale of the effective initialisation w.r.t. $\alpha$ when this latter is small enough. In fact, this lower bound on the integral of the loss along with a stronger assumption on the boundedness of the iterates lead to

$$\frac{\alpha_\infty}{\alpha} \underset{\alpha \to 0}{\lesssim} \left(\frac{\alpha^2}{\|\beta_{\ell_1}^*\|_1}\right)^\zeta, \tag{8}$$

for some $\zeta > 0$. Hence the smaller the initialisation scale $\alpha$ and the greater the benefit of SGD over GD in terms of implicit bias (see Appendix B.6 for more details).

Again, the proof of this proposition is technical and relies on considering appropriate Lyapunov functions which highly resemble to Bregman divergences, but which take into account the fact that the geometry changes over time. These overall decreasing Lyapunov's enable to bound the iterates as well as lower and upper bound the integral of the loss. The stochastic integrals which naturally appear are controlled with high probability using time-uniform concentration of martingales [19].

## 5 Experiments

### 5.1 Experimental setup for sparse regression

We consider the following sparse regression setup for our experiments. We choose $n = 40$, $d = 100$ and randomly generate a sparse model $\beta_{\ell_0}^*$ such that $\|\beta_{\ell_0}^*\|_0 = 5$. We generate the features as $x_i \sim \mathcal{N}(0, I)$ and the labels as $y_i = x_i^\top \beta_{\ell_0}^*$. SGD, GD and the SGF are always initialised using the same scale $\alpha > 0$ and it is specified each time. We use the same step size for GD and SGD and choose it to be the biggest as possible why still ensuring convergence. Note that since the true population covariance $\mathbb{E}[xx^\top]$ is equal to identity, the quantity $\|\beta_t - \beta_{\ell_0}^*\|_2^2$ corresponds to the validation loss.

### 5.2 Validation of the SDE model

In this section, we present an experimental validation of the stochastic gradient flow model. In Figure 3, for the same step size, we run: (i) the trajectory of gradient descent, (ii) 5 trajectories of stochastic gradient descent that correspond to different realisations of the uniform sampling over the data, (iii) 5 trajectories of the stochastic gradient flow (its Euler discretisation with $\mathrm{d}t = \gamma/10$)) corresponding to different realisations of the Brownian. We clearly see (left) that the loss behaves similarly for SGD and SGF across time. We also see that the validation losses (right) of the iterates of SGD and SGF have very similar behaviours. This tends to validate our continuous modelling from Section 2.2.

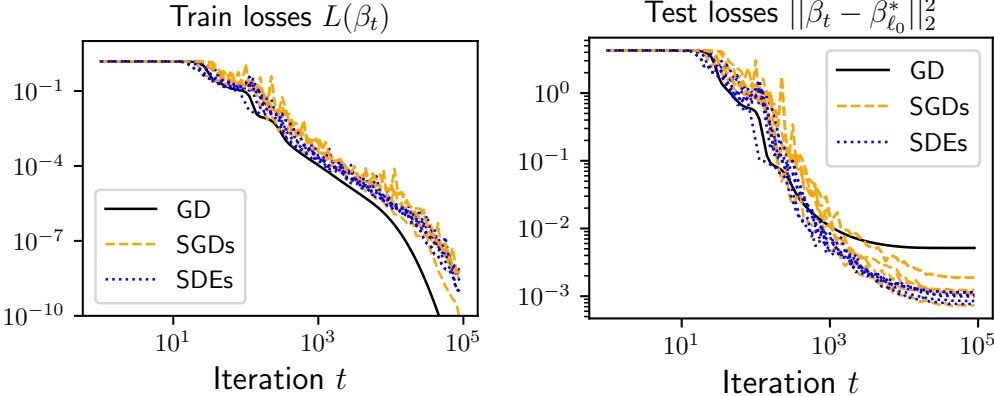

Figure 3: Sparse regression (see Section 5.1 for the detailed experimental setup). *Left and right*: the training and the validation losses behave very similarly, corroborating the continuous modelling.

### 5.3 GD and SGD have the same implicit bias, but from different initialisations.

In order to confirm and illustrate the main Theorem 1, we provide the following experiment which is illustrated Figure 4. We first run GD and SGD with the same step-size and initialise them both at $\alpha \mathbf{1}$ with $\alpha = 0.01$. As expected, the solution recovered by SGD generalises better. Then, using the iterates $\beta_t^{\mathrm{SGD}}$ from the first SGD run, we compute the value $\alpha_\infty = \alpha \exp(-2\gamma \operatorname{diag}(X^\top X/n) \int_0^\infty L(\beta_s^{\mathrm{SGD}})\mathrm{d}s) \in \mathbb{R}^d$ (the integral is approximated by its discrete time approximation with $\mathrm{d}t = \gamma$). We then run gradient descent but this time initialised at $w_{0,\pm} = \alpha_\infty$. According to our main result from Theorem 1, it should approximately (it would be exact if we ran SGF and GF) converge to the same solution as SGD initialised at $\alpha \mathbf{1}$. This is clearly observed Figure 4 (right). Also note that SGD and GD (initialised at $\alpha_\infty$) seem to have overall very

similar dynamics, this is not shown by our results and we leave this as future work. However keep in mind that though the validation losses converge at the same iteration rate, in terms of computation time, SGD is $n$ times faster.

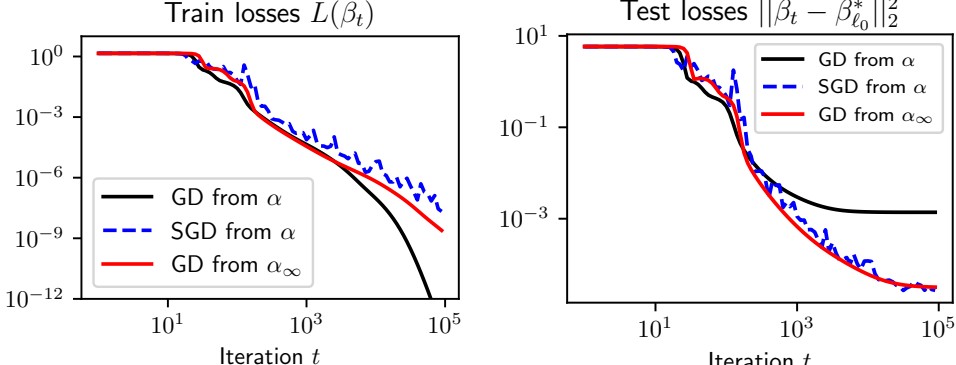

Figure 4: Sparse regression (see Section 5.1 for the detailed experimental setup). *Left and right*: SGD initialised at $\alpha\mathbf{1}$ converges towards the same point as GD initialised at $\alpha_\infty = \alpha \exp(-2\gamma \operatorname{diag}(X^\top X/n) \int_0^\infty L(\beta_s^{\mathrm{SGD}})\mathrm{d}s)$.

### 5.4 Doping the implicit bias with label noise

As largely discussed throughout the paper, the effect of the implicit bias is controlled by the convergence speed of the loss: the slower it converges, the sparser the selected solution will be. Hence the following question: can we leverage this knowledge to dope the implicit bias? We argue in this Section that the answer to this question is affirmative. Indeed, consider a sequence $(\delta_t)_{t\in\mathbb{N}} \in \mathbb{R}_+^\mathbb{N}$ and assume that we artificially inject some label noise $\Delta_t$ at time $t$, say for example $\Delta_t \sim \mathrm{Unif}\{2\delta_t, -2\delta_t\}$ (independently from $i_t$). This injected label noise perturbs the SGD recursion as follows:

$$w_{t+1,\pm} = w_{t,\pm} \mp \gamma \left(\langle \beta_w - \beta^*, x_{i_t}\rangle + \Delta_t\right) x_{i_t} \odot w_{t,+}, \qquad \text{where } i_t \sim \mathrm{Unif}(1, n). \quad (9)$$

As in Section 2.2, we can derive its related stochastic gradient flow (see Appendix D.1 for more details):

$$\mathrm{d}w_{t,\pm} = -\nabla_{w_\pm} L(w_t)\mathrm{d}t \pm 2\sqrt{\gamma n^{-1}(L(w_t) + \delta_t^2)} \, w_{t,+} \odot [X^\top \mathrm{d}B_t]. \quad (10)$$

Assuming that $(\delta_t)_{t\geq 0} \in (\mathbb{R}_+)^\mathbb{R}$ and $\gamma$ are such that the iterates converge, the corresponding implicit regularisation minimisation problem is preserved but with a "slowed down" loss: $\tilde{L}(\beta_t) := L(\beta_t) + \delta_t^2$ and the effective initialisation writes: $\tilde{\alpha}_\infty = \alpha \odot \exp\left(-2\gamma \operatorname{diag}(\frac{X^\top X}{n}) \int_0^{+\infty} \tilde{L}(\beta_s)\,\mathrm{d}s\right)$. The label noise therefore helps recovering a solution which has better sparsity properties. However, it must be kept in mind that adding too much label noise can significantly slow down the convergence of the validation loss or even prevent the iterates from converging. Yet, experimental results showing the impressive effect of label noise are provided Figure 5 in Appendix D.1.

## 6 Conclusion and Perspectives

In this paper, we have shown the benefit of using stochastic gradient descent over gradient descent for diagonal linear networks in terms of their implicit bias. Indeed, we prove that stochastic gradient flow acts as gradient flow but initialised at a smaller scale: this induces a sparser finale iterate. This effect is controlled by the speed of convergence of the loss. Moreover, we prove the convergence of the flow and exhibit an interesting link with mirror descent. Fully understanding this novel type of dynamics could help to grasp the implicit biasing properties of stochastic gradient descent in other frameworks. It is also natural to ask whether the integral of the loss also controls the difference of implicit regularisation for more general architectures. It would also be interesting to analyse how this property adapts to log losses known to lead to max-margin solutions in classification.

**Acknowledgements.** NF would like to thank Nathan Srebro for introducing him to the question of SGD's implicit bias as well as for the stimulating discussions they had during his visit at EPFL.

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
