# OpenReview forum: "Implicit Bias of SGD for Diagonal Linear Networks: a Provable Benefit of Stochasticity"
_NeurIPS.cc/2021/Conference — NeurIPS 2021 Poster_

### Official Review · Reviewer_9yUy · 2021-06-27

**Rating:** 7
**Confidence:** 4

**Summary:**

In this paper the authors study the dynamics of SGD for diagonal linear networks. The exact expression of implicit bias is derived, showing that SGD with some initialization scale is equivalent to full batch GD (or gradient flow) will a smaller effective initialization scale. In addition, SGD slows down the convergence speed, and the slower the convergence, the smaller the effective initialization scale. For sparse problems, the smaller effective initialization scale induces better generalization, as also demonstrated empirically on synthetic data.

**Main Review:**

Overall, I think this is a good paper. While previous works that study diagonal linear networks only consider the case of full batch GD (or gradient flow), the authors provide a novel analysis for the case of SGD. To this end, the authors model the dynamics through a stochastic differentiable equation, which solution is then analyzed.

The analysis is highly non-trivial and this is an important step forward since in practice SGD is usually used. Also, the effect of label noise on the effective initialization scale is naturally integrated into the proposed analysis.

The paper is well written and easy to read.

Weaknesses:
1.	The analysis only considers the case of batch size = 1 and it is not clear how the batch size affects the effective initialization scale.
2.	The effect of the step size in Theorem 1 is not clearly discussed. On the one hand, a large step size will accelerate the convergence and thus the integral of the loss will be small, but on the other hand, the step size appears explicitly in the exponent.  So, is increasing step size result in smaller effective $\alpha$ ? If yes, is it correct also for GD or only for SGD ?
3.	It would be good to complement the paper will experiments on real data with real networks. Clearly, the interaction of initialization scale, batch size and step size is an important practical question.

More comments:
- line 130: I don’t think the model is equivalent to the model $\beta=u\odot v$, as discussed in section 4 of [1].
- In Theorem 1 the tradeoff in $p$ is not clear. It seems that $p$ can be arbitrarily small.
- In Theorem 1, can you explain why a bound on the step size is needed ? Is it for stability reasons ?
- Several relevant papers are missing:

On the implicit bias of SGD:

https://arxiv.org/abs/2101.12176

https://arxiv.org/abs/2003.07802

On diagonal linear networks and mirror descent:

https://arxiv.org/abs/2007.06738

https://arxiv.org/abs/2004.01025

- In section 3 the kernel regime is discussed. Is it possible to have a similar discussion for the rich regime ? Is there a difference between SGD and GD in the rich regime ?
- In line 220 the integral goes to 0, I couldn’t find the proof in the appendix. Also, the upper bound of the integral in proposition 3 goes to infinity when $\alpha\rightarrow\infty$. Can you explain this ?
- Can you explain why the equality in line 303 is correct ?
- $\alpha$ is sometimes used as a scalar and sometimes as a vector, and it will be good to distinguish between them, e.g. make vector a bold symbol. For example, in proposition 3 it is not clear if $\alpha$ is a scalar or a vector.
- In section 5.1 please specify what is the step size.
- In section 5.3 will be useful to compare the result to [12].

typos:
- line 160: I think the second diag is not needed.
- line 316: why -> while
- line 348: understand -> understanding
- line 444: $\alpha_t$ is incorrectly defined

**Time Spent Reviewing:**

15

---

> ### Author Response · Authors · 2021-08-10
> **Review answer**
>
> Thank you very much for the time you took to review the paper.
>
> **Role of minibatch size.** Though we did not mention it, our analysis can easily be extended to a batch size larger than $1$: in terms of implicit bias, SGD with step size $\gamma$ is equivalent to mini-batch (with replacement) SGD with batch size $b$ and step-size $\gamma b$. Indeed, using a mini-batch sampled with replacement of size $b$ only changes the noise covariance up to multiplicative constant as: $Cov_{i_t}(\xi_{i_t}^b (\beta)) = \frac{1}{b} Cov_{i_t}(\xi_{i_t}^{b = 1} (\beta))  $. The associated SDE, for a step size $\gamma$,  is therefore $ \mathrm{d} w_{t,\pm} = - \nabla_{w_\pm} L(w_t) \mathrm{d} t \pm  \sqrt{\gamma b^{-1} n^{-1} L (w_t) }\  w_{t,+} \odot [X^\top \mathrm{d}  B_t]$. Hence, the rest of the analysis still holds but with an effective step size $\gamma_{\mathrm{eff}} = \gamma / b$ (hence larger step-sizes can be used which still ensure convergence, as expected). The exact same reasoning can be done for mini-batch without replacement and our analysis would hold this time with: $\gamma_{\mathrm{eff}} = \gamma (n - b)  / (n - 1) b$. This is an interesting observation and it will be added in the revised version.
>
> **Role of step size.** This is a very important observation we only discuss in the appendix. We will add this discussion in the main paper. We show in Lemma 7 in the appendix (note that there is typo in the statement of the lemma line 614, the equality on step size $\gamma$ should in fact be *an upperbound*) that the quantity $\gamma \int_0^\infty L(\beta_s) \mathrm{d} s$ is lower bounded by a quantity which is *strictly increasing* with $\gamma$. This lemma therefore recommends picking the largest $\gamma$ possible (as long as there is convergence, see also lines 610-613 in the appendix). Though we did not show it, this observation is probably also true for GD (however the implicit bias of GD does not a priori depend on this quantity contrary to SGD).
>
> **Experiments with real data and real networks.** The main focus of the paper being theoretical (along with some simple toy examples illustrating the validity of the results), we prefered keeping more in depth experiments on real data with real networks for future work.
>
> **Link with the $\beta = u \odot v$ model.** The link with the diagonal linear network is explicitly made through the linear mapping $(u, v) = (w_+ + w_-, w_+ - w_-)$ (note that there are $2 d$ parameters in this parametrisation). In section 4 of [1] they consider a different parametrisation which is $\beta = u_+ \odot v_+ - u_- \odot v_- $ (there are $4 d$ parameters here).
>
> **Can $p$ be arbitrarily small?** The tradeoff appears in the maximal value of $\gamma$ which ensures convergence (with probability $1 - p$): i.e. $p$ can be taken arbitrarily small, however from the upperbound on $\gamma$ given line 226, the step size must then be taken arbitrarily small. We will make this clearer in the revised version.
>
> **Kernel and rich regimes.** Even if we briefly mention the kernel regime, the benefit of SGD over GD is truly relevant in the rich regime where $\int_0^\infty L(\beta_s) \mathrm{d}s$ is not too small (lines 209-211).
> In the kernel regime, when $\alpha$ goes to $\infty$, as mentioned in line 220, we expect that $\int_0^\infty L(\beta_s) \mathrm{d}s \to 0$.
> In the deterministic case it is easy to show that it is true (through the equality in line 303), however in the stochastic case it is more tricky to show and we do not have a proof at the moment (we unfortunately forgot to remove the sentence line 220 from the submission). We thank the reviewer for pointing this oversight out and we will either find a proof for the revised version or remove this statement.
>
>
> **Line 303.** We apologize for not having referenced where to find the proof. It corresponds to lines 665-666 in the appendix: $\frac{\mathrm{d}}{ \mathrm{d} t} D_{\Psi}(\beta_\infty, \beta_t) = - \langle \nabla L(\beta_t) , \beta_t - \beta_\infty \rangle  = - 2 L(\beta_t)$, where the last equality is because we consider the square loss. Integrating from $0$ to $+\infty$ leads to $\int_0^\infty L(\beta_s) \mathrm{d} s = \frac{1}{2} D_{\Psi}(\beta_\infty, \beta_0)$ (the $1 / 2$ factor is missing in the main text).
>
> **Missing references.** Thank you for all the relevant references you gave us and all the suggestions and precisions you asked for: they will be of great help to improve the clarity of the paper.

---

> > ### Comment · Reviewer_9yUy · 2021-08-24
> > **Response**
> >
> > Thank you for the detailed response.
> >
> > One unresolved comment:
> > In Theorem 1, can you explain why a bound on the step size is needed (line 197) ? Is it for stability reasons ?
> >
> > I also read all other reviews and decided to keep my score.

---

> > > ### Author Response · Authors · 2021-08-25
> > > **Upper bound on the step size**
> > >
> > > Thank you for your response and sorry we forgot to answer this comment in our original response. As you mention the upper bound on the step size is expected and is required for stability reasons.  Indeed we cannot expect the iterates to converge for arbitrarily large step sizes (for the same reasons as SGD: otherwise the iterates will diverge).  Also note that the bound we give on the step size (line 226) is macroscopic and depends on relevant parameters of our problem.

---

### Official Review · Reviewer_MruN · 2021-07-07

**Rating:** 7
**Confidence:** 2

**Summary:**

Summary:
This paper studies the implicit bias of stochastic gradient flow (SGF) over a linear model, re-parameterized as a quadratic difference: $\beta=(w_1)^2 - (w_2)^2$. The paper establishes that in the said setting, SGF has implicit bias towards sparsity (lower $\ell_1$ norm w.r.t regular GF).

**Ethical Concerns:**

No ethical concerns.

**Limitations And Societal Impact:**

Limitations:
- Result applies only to diagonal networks.
- Initialization schemes does not follow standard schemes.

The reviewer does not see any negative social impact of the submitted work.


**Main Review:**

**Originality** - The paper uses original analysis to discuss stochasticity in the specific framework. The paper insinuates that SGF is presented in this work (line 151-154). Yet previous work exists [2] therefore it is difficult to determine exactly how novel are the ideas. In addition, the paper discusses some related work rather superficially (lines 75-77) neglecting to cite relevant works such as [1] that discuss implicit bias of linear regression with SGD vs GD.

**Quality** - Overall, the paper is well written. My biggest concern is the setting, it is not clear to me why diagonal networks are an interesting re-parameterization of linear networks. Additional issues are (1) Unrealistic initialization, the analysis holds for any initialization such that $w_{+}=w_{-}$ which also means $\beta_{w}=0$, it is not clear how this relates to standard initialization schemes. (2) The role of $\alpha$ - as stated in the paper, the tradeoff is between generalization and optimization (line 212), therefore a user can set it to a desired value and control the tradeoff, it is not clear how the $\alpha_{\infty}$ relates to initial $\alpha$ except from being upper bounded by it. Further discussion can be interesting.

**Clarity** - The paper is very clear in an abstract level, the claims in the main manuscript as well as the specific setting are very clearly written. As for the analysis, there seems to be no apparent structure for the mathematical analysis. In the appendix it is not clear what are the important details and which sections are more technical. It made it hard to understand specific claims. For example, I would expect a more clear presentation and discussion of $\alpha_{\infty}$ which is written implicitly in proposition 1.

**Significance** - Understanding the effect of stochasticity on the implicit bias of GD is extremely significant, with that being said, this work is limited to a non-realistic setting which makes this result weaker.

[1] Wu, Jingfeng and Zou, Difan and Braverman, Vladimir and Gu, Quanquan. Direction Matters: On the Implicit Bias of Stochastic Gradient Descent with Moderate Learning Rate

[2] Ali, Alnur and Dobriban, Edgar and Tibshirani, Ryan. The Implicit Regularization of Stochastic Gradient Flow for Least Squares


**---------------------------------------------------------------------------**

**Update**: Thank you for the detailed rebuttal.

After reading other reviews and authors' responses - I believe I underestimated the significant of the problem setup and the contribution of the paper. Therefore I chose to change my score accordingly.

Good luck.



**Time Spent Reviewing:**

15

---

> ### Author Response · Authors · 2021-08-10
> **Review answer**
>
> Thank you very much for the time you took to review the paper.
>
> **Novelty in the SGF modelling.** It is worth noting that we never claimed that modelling SGD with a SDE is a new contribution. We refer to our related work for this. The only *novelty* is that we model a *specific* non-linear model by a *specific* SDE and we believe that our model correctly mirrors the behavior of the noise coming from SGD. Note in comparison that the article [2] you mentioned is on the linear case (see below).
>
> **Comparison with previous works [1], [2].** The two references you gave [1] and [2] are very relevant and we thank you for pointing out these two works. However, note a crucial difference: these two articles present results on implicit bias for *linear* models. On the contrary, a very important aspect of our work is to crucially provide a non-linear setting where the implicit bias of SGD is fairly non-trivial and different from the one of GD.
>
> **Motivation for the diagonal networks.** We believe that when tackling a not-yet-understood phenomena (here: the implicit bias of SGD), we must first fully understand it in the simplest model that exhibits it. Dealing with more sophisticated / realistic architectures is an obvious direction for future work. However it is mandatory to be able to carry out the analysis in simple settings before trying to tackle more complicated ones.
>
> **Unrealistic initialisation.** We agree with the reviewer that the initialisation taken is quite restrictive. Once again, the aim of our work is to focus on the effect of the stochasticity of SGD on the implicit bias, hence non-usual initialisations like $w_+ = w_-$ were not our primary concern but allowed us to simplify the analysis and moreover to give clear interpretations. Note that the analysis made in the recent paper [On the Implicit Bias of Initialization Shape: Beyond Infinitesimal Mirror Descent, Azulay et al, 2021] should help us to extend our results to the case of general initialisation, but at the expense of less readable results. Yet, we agree that this would be an interesting future direction.
>
> **Link between $\alpha$ and $\alpha_\infty$.** Indeed, in practice, one chooses an $\alpha$ which is not too small in order for the optimisation not to take too long but which still ensures generalisation. However, we prove that for the same initialisation, SGD converges to a solution that will generalise better than GD (since $\alpha_\infty < \alpha$).

---

### Official Review · Reviewer_yevU · 2021-07-15

**Rating:** 7
**Confidence:** 4

**Summary:**

The paper studies the implicit bias of SGD in diagonal linear networks. Specifically, through the study of SGD continues time version, i.e., stochastic gradient flow, the authors investigate the noise influence on the generalization performance. The authors show that with high probability stochastic gradient flow converges to zero loss solution that enjoys better generalization properties than the solution obtained by gradient flow. In addition, they show that slower convergence results in better generalization.

**Ethical Concerns:**

None.

**Limitations And Societal Impact:**

The authors discussed the limitations of their results.

**Main Review:**

The paper studies the implicit bias of SGD in diagonal linear networks and the square loss. To this end, the authors model SGD dynamics using its continuous time, i.e., stochastic gradient flow (SGF).

The paper main contributions are:
-	The authors prove that with high probability SGF converges to zero loss solution.
-	The authors characterize the obtained solution (similarly to [32]). Specifically, their characterization implies that SGF biases the flow toward solutions with better sparsity inducing properties than GF (smaller effective $\alpha$).
-	The authors provide experimental results that validate their model and theoretical results.

My main question:
-	When modeling SGD the authors assume that all partial losses are approximately uniformly equal to their mean (Appendix A). Can you please elaborate and justify this assumption? Also, this assumption should appear in the main paper.

Clarity/writing:
-	The paper is well written, and the results are clearly stated.

Minor comment:
-	Figure 2 will be more clear if you include the fact that $\Vert \beta_t - \beta_{\ell_1}^*\Vert_2^2$ corresponds to the validation error in the caption (and not just in section 5.1)

---

Update: Thank you for the detailed rebuttal. I accept your answer regarding the modeling choices you made.


**Time Spent Reviewing:**

8

---

> ### Author Response · Authors · 2021-08-10
> **Review answer**
>
> Thank you for your appreciation of this work and the time you took to review the paper.
>
> **Partial losses approximation for the SDE model.** As mentioned in the main paper (lines 159-164) and in the appendix, this simplification is a model rather than an assumption. We agree with the remark of the reviewer and do not claim that we can justify this assumption (similarly to the fact that we cannot justify the modelling of SGD by the SGF). We presented this model for the sake of clarity: it simplifies the analysis as well as the interpretations. However, note that in Appendix E.1 we give some results in the case where we do not assume that all partial losses are uniformly equal to their mean: this does not modify the reasoning and the nature of our results.
>
> **Validation error.** We should have clearly explained that $\Vert \beta_t - \beta^*_{\ell_1} \Vert_2^2$ corresponds to the validation error. We will make it clearer in the revised version.

---

### Official Review · Reviewer_a887 · 2021-07-15

**Rating:** 7
**Confidence:** 3

**Summary:**

This paper studies the implicit bias of SGD over GD trained solutions for a regression task, with MSE loss, and a quadratic parameterization of the regression coefficients.  Building off recent analysis demonstrating how the initialization for GD toggles between an implicit sparse/L2 regularization on the interpolating solution, they demonstrate that SGD has the same effect but with a strictly smaller initialization (leaning to sparse).  To get this conclusion they model SGD using an SDE with a spatially dependent, anisotropic diffusion.  Lastly, they demonstrate their main claims on synthetic data.

**Limitations And Societal Impact:**

Yes.

**Main Review:**

**Originality/Significance:**
The key qualitative assumption discussed in this paper is "the effect of the implicit bias is controlled by the convergence speed of the loss: the slower it converges, the sparser the selected solution will be."  As far as I am aware, this is a very novel statement and quite significant (even if only made for a constrained class of models).  Overall, this work was very well done, however I think more emphasis could have been on experiments (such as some of the suggestions I give below) and discussing the qualitative takeaways that might apply to the general class of nonlinear networks. I agree with the authors, the most interesting direction for future work would be "whether the integral of the loss also controls the difference of implicit regularisation for more general architectures."  A longer discussion at the end on the connections of this analysis to nonlinear networks (where the benefits of SGD really shine compared to GD) would be very helpful.

**Clarity/Quality:**
The writing and mathematics in this paper are incredibly clear and well done.  I found the flow very natural and the discussion of related work very helpful for grounding their analysis.  The experiments were well documented and explained.

**Areas for Improvement:**
- Throughout the work (such as in abstract) you say "better the bias", but its not clear what "better" means.  Is there another description that might be more specific you can use as your tag line? Such as "sparse"?
- This analysis is done assuming a batch size of 1? Can this be extended to mini-batches?  A discussion about this assumption/limitation should be given.
- In the buildup to the SDEs you never explicitly assume the noise term is Gaussian, as done in many of the "SDE for SGD" papers.  Yet implicitly it seems like you are,  as the Euler-Maruyama discretization of equation (3) would lead to gaussian noise with the same covariance as your noise term.  I think its worth having a discussion about this, as the Gaussian assumption has been one of the major sources for complaint on the "SDE for SGD" literature?
- The statement on line 206/207 "the recovered solution minimizes the same potential as for gradient flow but with an effective parameter which is strictly smaller" makes me want to see an experiment where you run SGD with $\alpha$ and compute $\alpha_\infty$ from trajectory then run GD with $\alpha_\infty$ and compare overall optimization time and final solution.  An experiment like this would definitely help convince me of the statement "However it is clear experimentally that SGD systematically largely wins the trade-off over GD".  If for example the SGD and GD trajectory both converge to same performing solution but SGD does it faster, then I would be very convinced on this.
- All your experiments used features generated from an isotropic gaussian. I understand that this was done to make distance to sparse solution a measure of validation performance, but I would have been very curious to see how dynamics change with features from an anisotropic gaussian.  Intuitively, I would assume that anisotropy would have a large effect on the SGD dynamics...and maybe even the final solution reached.
- The section "Doping the implicit bias with label noise" is interesting.  I would devote more space to this discussion in revision and pull experiments up from appendix.

**Specific Comments:**
- Consider citing the recent work "ON THE ORIGIN OF IMPLICIT REGULARIZATION IN STOCHASTIC GRADIENT DESCENT"
- [line 105-106] "All inequalities between vectors should be understood value by value" => replace "value by value" with "element-wise"?
- In the section "Implicit bias and hyperbolic entropy" the analysis in [32] you are referencing is for the same quadratic parametrization correct?  If so, can you make this clear.

**Time Spent Reviewing:**

5

---

> ### Author Response · Authors · 2021-08-10
> **Review answer**
>
> Thank you for the time you spent reading and commenting on our paper. We are grateful and this will help us clarify and improve the paper.
>
> **What is a *better* bias.** We agree that the wording  “better bias” is unclear. By “better”, we mean that the iterates are biased towards an interpolating solution which enjoys *sparsifying* properties and which therefore leads to better generalisation for classical sparse linear models. We will make this clearer in the revised version by avoiding such vague expressions.
>
> **Role of minibatch size.** Though we did not mention it, our analysis can easily be extended to a batch size larger than $1$: in terms of implicit bias, SGD with step size $\gamma$ is equivalent to mini-batch (with replacement) SGD with batch size $b$ and step-size $\gamma b$. Indeed, using a mini-batch sampled with replacement of size $b$ only changes the noise covariance up to multiplicative constant as: $Cov_{i_t} (\xi_{i_t}^b (\beta)) = \frac{1}{b} Cov_{i_t} (\xi_{i_t}^{b = 1} (\beta)) $ . The associated SDE, for a step size $\gamma$,  is therefore $ \mathrm{d} w_{t,\pm} = - \nabla_{w_\pm} L(w_t) \mathrm{d} t \pm  \sqrt{\gamma b^{-1} n^{-1} L (w_t) }\  w_{t,+} \odot [X^\top \mathrm{d}  B_t] $. Hence, the rest of the analysis still holds but with an effective step size $\gamma_{\mathrm{eff}} = \gamma / b$ (hence larger step-sizes can be used which still ensure convergence, as expected). The exact same reasoning can be done for mini-batch without replacement and our analysis would hold this time with: $\gamma_{\mathrm{eff}} = \gamma (n - b)  / (n - 1) b$ . This is an interesting observation and it will be added in the revised version.
>
> **Gaussian assumption on the noise ?** We agree with the reviewer that properly modelling SGD by an SDE implicitly demands some assumptions on the noise. However, they are not very stringent since consistency of this modelling in the small step size limit only requires the noise to have a *finite variance* (see *Modeling with Itô Stochastic Differential Equations*, Allen E, 2007, Springer). Even if this finite variance assumption may not be true in some extreme cases (see particularly the works of Simsekli et al. on heavy tails), a large class of processes still satisfies it without being Gaussian. Finally, we would like to emphasise that we simply provide *a model* and *do not make any claim* on the validity of the SDE modelling for SGD (but provide some experimental evidence that they behave similarly).
>
> **Comparison of SGD initialised at $\alpha$ and GD initialised at $\alpha_\infty$.** It is a very interesting question which we have considered: we present such an experiment in Figure 5 in the appendix. Note that experimentally the iteration complexity of SGD and GD are similar but SGD is still *$n$ times faster* in terms of oracle complexity and hence much faster.
>
> **Effect of the anisotropy of Gaussian features.** Though we did not include them in the final version, we did perform experiments with features generated from non-isotropic gaussians. Doing so changes the final solution, however the gap between SGD and GD is similar. For completeness we will add these experiments in the revised version.
>
> **Other comments.** Thank you for all the other comments. If the paper is accepted, we will make use of the additional page in order to include the experiments concerning label noise in the main paper and refine our related work by adding the pertinent reference mentioned on the implicit regularisation of SGD.

---

> > ### Comment · Reviewer_a887 · 2021-08-12
> > **Thank you for the response**
> >
> > Thank you for the response.  You have answered most of my questions. I think this work is a very nice demonstration of a simple setting where the interaction of initialization, stochasticity, and generalization can be understood theoretically.  Of course, the limitations are that this is a simplified setting, is somewhat incremental analysis beyond what is already introduced in the literature on diagonal networks, and the experimental section is quite limited. After reading your rebuttal and other reviews, and based on these limitations, I will maintain the rating I originally gave of 7/10.
> >
> > For the revision I would strongly recommend:
> > - Including the discussion on minibatch in the body (as echoed by reviewer 9yUy) or in fact just doing all analysis with a batch size $b$.
> > - Move figure 5 in the appendix up to where figure 2 is currently in the body.
> > - Extend related work section, with emphasis on discussing how your analysis relates to other diagonal network literature such as [32] and other works using SDEs to study SGD.
> > - Devote a longer discussion to future work and extensions.

---

### Decision · Program_Chairs · 2021-09-27

**Decision:**

Accept (Poster)

**Comment:**

Reviewers generally agreed that the paper is clearly written, technically solid and interesting.  Reservations with regards to the significance of the analyzed setting (diagonal linear neural networks) were raised, but given that this setting already received notable attention, the committee reached a decision by which this paper comprises a meaningful step forward (analysis of stochasticity), and should thus be accepted to the conference.